# Proper Learnability and the Role of Unlabeled Data

**Julian Asilis**                                                    ASILIS@USC.EDU
**Siddartha Devic**                                                  DEVIC@USC.EDU
**Shaddin Dughmi**                                                   SHADDIN@USC.EDU
**Vatsal Sharan**                                                    VSHARAN@USC.EDU
**Shang-Hua Teng**                                                   SHANGHUA@USC.EDU
*University of Southern California*

**Editors:** Gautam Kamath and Po-Ling Loh

## Abstract

Proper learning refers to the setting in which learners must emit predictors in the underlying hypothesis class $\mathcal{H}$, and often leads to learners with simple algorithmic forms (e.g., empirical risk minimization (ERM), structural risk minimization (SRM)). The limitation of proper learning, however, is that there exist problems which can only be learned improperly, e.g. in multiclass classification. Thus, we ask: Under what assumptions on the hypothesis class or the information provided to the learner is a problem properly learnable? We first demonstrate that when the unlabeled data distribution is given, there always exists an optimal proper learner governed by *distributional regularization*, a randomized generalization of regularization. We refer to this setting as the *distribution-fixed* PAC model, and continue to evaluate the learner on its worst-case performance over all distributions. Our result holds for all metric loss functions and any finite learning problem (with no dependence on its size). Further, we demonstrate that sample complexities in the distribution-fixed PAC model can shrink by only a logarithmic factor from the classic PAC model, strongly refuting the role of unlabeled data in PAC learning (from a worst-case perspective).

We complement this with impossibility results which obstruct any characterization of proper learnability in the classic (realizable) PAC model. First, we observe that there are problems whose proper learnability is logically *undecidable*, i.e., independent of the ZFC axioms. We then show that proper learnability is not a monotone property of the underlying hypothesis class, and that it is not a *local* property (in a precise sense). We also point out how the non-monotonicity of proper learning obstructs relaxations of the distribution-fixed model that preserve proper learnability, including natural notions of class-conditional learning of the unlabeled data distribution. Our impossibility results all hold even for the fundamental setting of multiclass classification, and go through a reduction of EMX learning (Ben-David et al., 2019) to proper classification which may be of independent interest.

**Keywords:** PAC learning, proper learning, semi-supervised learning, classification, regularization.

## 1. Introduction

We are motivated by the following fundamental question in computational learning theory.

*When are supervised learning problems properly learnable?*
*If so, by what kinds of proper learners?*

Classification stands as perhaps the most fundamental setting in supervised learning. Namely, a learner receives a sequence of training points — consisting of datapoints and their accompanying labels — and must learn to correctly predict the label of an unseen datapoint. Notably, a predicted

label is either correct and incurs a loss of zero or is incorrect and incurs a loss of one; there is no notion of a near-miss. Typical applications of this framework include image and document classification, sentiment analysis, and facial recognition, to name a few. Binary classification, the setting in which there are only two possible labels, is perhaps the single most-studied regime of learning. In this setting, the celebrated fundamental theorem of statistical learning theory establishes that a hypothesis class $\mathcal{H}$ is learnable precisely when its VC dimension is finite, in which case it can be learned nearly-optimally by the learning rule of empirical risk minimization (ERM). Recall that ERM, upon receiving a training set $S$, simply selects one of the hypotheses in $\mathcal{H}$ with best fit to $S$. Notably, ERM learners are an instance of *proper learning*, that is, learning under the constraint that the emitted predictor always be an element of the underlying class $\mathcal{H}$.

*Multiclass classification* proceeds identically as in binary classification, save for the fact that the collection of possible labels for the data — denoted $\mathcal{Y}$ — is permitted to be arbitrarily large, perhaps even infinite. To what extent do insights from binary classification extend to the multiclass case? Perhaps less than one may expect. Daniely and Shalev-Shwartz (2014) showed that there exist multiclass classification problems which are learnable yet cannot be learned by *any* proper learner. Notably, this demonstrates that ERM, perhaps the quintessential workhorse of machine learning (and particularly binary classification), does not enjoy the same success in multiclass classification.

In fact, the task of characterizing multiclass learnability via some choice of dimension, analogous to the VC dimension for binary classification, remained a major open problem for decades. Brukhim et al. (2022) recently demonstrated in a breakthrough result that learnability is in fact characterized by the *Daniely-Shalev-Shwartz* (DS) dimension. Further, they exhibited learners for all classes of finite DS dimension, based upon certain extensions of the *one-inclusion graph* predictor of Haussler et al. (1994) and several novel ideas such as *list* PAC learning. Interestingly, the learner of Brukhim et al. (2022) employs techniques which are strikingly different from — and more intricate than — ERM and the standard algorithmic approaches of binary classification. The complexity of existing (improper) learners for multiclass classification, and the necessity of some such complexity by the result of Daniely and Shalev-Shwartz (2014), raises a natural question: Can one succeed with simpler learning rules, under additional assumptions on $\mathcal{H}$ or on the learning model?

First, we observe that when a learner can infer a high degree of information about the marginal distribution over unlabeled datapoints, $\mathcal{D}$, then improper learnability is equivalent to learnability by a proper learner. (As can be seen by a simple use of the triangle inequality to "de-improperize" any improper learner by rounding its outputs to their nearest hypotheses in $\mathcal{H}$.) More strikingly, we establish that there always exists one such learner based upon a *distributional regularization*, a form of regularization which assigns a score to each distribution over hypotheses in $\mathcal{H}$ (i.e., to each randomized hypothesis). These results are formalized using the notion of *distribution-fixed* PAC learning, in which the learner receives both a training sample $S$ and the marginal distribution $\mathcal{D}$. We show an equivalence between learnability in the PAC and distribution-fixed PAC setting, for any bounded metric loss, along with an approximate equivalence between sample complexities. Perhaps surprisingly, therefore, knowing the marginal does not change learnability — or even considerably alter sample complexities, in the worst case — but rather greatly simplifies the form of the optimal algorithm. We ask whether proper learnability is equivalent to learnability by regularization (i.e., by *structural risk minimization* (SRM)) in the classic PAC model as well, though leave that question open.

We complement this with several impossibility results demonstrating that the landscape of proper multiclass learning is considerably more complex than that of improper learning. First, we

show that proper learnability can be logically *undecidable*, i.e., independent of the standard ZFC axioms. This implies that it is not provable whether certain classes $\mathcal{H}$ are properly learnable or not. Secondly, we show that proper learnability is not a local property: there exist classes $\mathcal{H}, \mathcal{H}'$ such that $\mathcal{H}|_S = \mathcal{H}'|_S$ for every finite set $S$ of unlabeled datapoints, yet $\mathcal{H}$ is properly learnable and $\mathcal{H}'$ is not. Lastly, we demonstrate that proper learnability is not a *monotone property* — it is not invariant under taking subsets or supersets. This poses several obstructions to characterizing proper multiclass learnability, and demonstrates that any such characterization must differ fundamentally from the usual dimensions enjoyed by learning theory (e.g., VC, DS, etc.).

In light of our positive result, it is natural to ask whether one can draw a connection between proper learnability and unsupervised learning (i.e., the ability to infer distributional information about the unlabeled data distribution $\mathcal{D}$) in the classic PAC model. Perhaps $\mathcal{H}$ is properly learnable precisely when $\mathcal{D}$ can be learned in some "class-conditional" sense which depends upon $\mathcal{H}$? Several such conditions have been proposed by Hopkins et al. (2023) to study binary classification, including Weak TV-learning, Strong TV-learning, and Exact TV-learning. All such definitions are monotone, however, and thus — by our previous impossibility result — cannot characterize proper learnability. More generally, any notion of learning the marginal in a class-condition manner will likely take the form of a monotone property, and thus fail to characterize proper learnability. In short, a precise characterization of proper learnability may require a fundamentally different approach than the standard techniques of (improper) supervised learning.

## 1.1. Related Work

**Proper learnability.** We focus primarily — but not exclusively — on the setting of multiclass classification, i.e., learning under the 0-1 loss function. When $|\mathcal{Y}| = 2$, one recovers binary classification, for which learnability is characterized by the VC dimension and empirical risk minimization (ERM) is a nearly-optimal learner (Blumer et al., 1989; Shalev-Shwartz and Ben-David, 2014). As ERM is proper, learnability is thus equivalent to proper learnability in the binary case.[1] In the multiclass case, $\mathcal{Y}$ is permitted to be of arbitrarily large size (even infinite), and the equivalence between proper learnability and improper learnability from the binary case was shown to fail by Daniely and Shalev-Shwartz (2014). They also proposed the *Daniely-Shalev-Shwartz* (DS) *dimension*, and conjectured that it characterizes improper multiclass learnability. This was recently confirmed in a breakthrough result of Brukhim et al. (2022), resolving a long-standing open question. Regarding algorithmic templates for multiclass learning, relatively little is known: Brukhim et al. (2022) designed one learner for general DS classes, using an intricate sequence of arguments and algorithmic techniques (e.g., *list* PAC learning, sample compression, one-inclusion graphs, etc.). It is natural to ask for simpler learners than that of Brukhim et al. (2022), perhaps which bear a closer resemblance to algorithms enjoying practical success (e.g., structural risk minimization (SRM)). Recently, Asilis et al. (2024b) made some progress by demonstrating that there always exist optimal learners taking the form of *unsupervised local regularization*, a certain relaxation of classical regularization. The proof is non-constructive, however, saying little about the precise *form* of the regularizer or the learner. Perhaps most relevant to our work is the line of research studying learnability via ERM.

---

1. Attaining the optimal sample complexity, however, is known to require improper learning in general. Interestingly, recent work has demonstrated that the improperness requirement for optimal learning can be satisfied using simple aggregations of proper learners, such as a majority of only 3 ERM learners (Hanneke, 2016; Larsen, 2023; Aden-Ali et al., 2024). In the multiclass setting, however, there are learnable classes which cannot be learned by any aggregation of a finite number of proper learners (Asilis et al., 2025).

This includes work demonstrating that there can be arbitrarily large gaps between the sample complexities of different ERM learners, and that the sample complexity of ERM is closely related to the *graph dimension* (Daniely et al., 2015). Learnability by any ERM learner is *not* equivalent to proper learnability, however, and thus this does not directly address our primary question. Regarding the issue of *optimal* proper learning, Bousquet et al. (2020) studied the conditions under which a binary hypothesis class $\mathcal{H}$ can be learned with optimal sample complexity by a proper learner, and established a characterization via finiteness of the *dual Helly number*, under general conditions on $\mathcal{H}$.

**The role of unlabeled data.** There is a long line of work studying the power of unlabeled data in learning, often formalized by the setting in which a learner receives both labeled and unlabeled datapoints, i.e., *semi-supervised learning* (SSL) (Kääriäinen, 2005; Zhu, 2005; Chapelle et al., 2006; Van Engelen and Hoos, 2020). One direction has studied SSL under the assumption that there is a relationship between the unlabeled data distribution $\mathcal{D}$ and the true labeling function $h^*$, and demonstrated results supporting the power of unlabeled data in this setting (Castelli and Cover, 1995; Seeger, 2000; Rigollet, 2007; Singh et al., 2008; Niyogi, 2013). Another line of work demonstrates that in that absence of any such assumptions, unlabeled data has little effect in binary classification from a worst-case perspective (Ben-David et al., 2008; Darnstädt and Simon, 2011; Göpfert et al., 2019). Yet another direction of work studies the power of unlabeled training points from a fine-grained perspective, examining learners' sample complexities on particular data distributions rather than on a worst-case basis, and establishes the value of unlabeled data in learning binary classes of infinite VC dimension (Darnstädt et al., 2013). We study a setting which makes no assumption on the unlabeled distribution $\mathcal{D}$ or the true labeling function $h^* \in \mathcal{H}$, but assumes that the learner receives complete information of $\mathcal{D}$. The learner is then judged on a worst-case basis over all possible (realizable) distributions. This most closely aligns with the "utopian" model of SSL studied by Ben-David et al. (2008) and Lu (2009). Notably, Göpfert et al. (2019) demonstrated that this setting — which they refer to as simply "knowing the marginal" — is of no additional help for binary classification. (I.e., the worst-case expected error rate of a learner does not improve by granting it knowledge of the marginal.) Our analogous result can be seen as extending this finding to a broader collection of bounded metric loss functions (Theorem 9).

**Decidability in learning.** In Section 4, we establish several obstructions to characterizing proper learnability in multiclass classification, including by demonstrating that there exist classes $\mathcal{H}$ for which it is *logically undecidable* whether $\mathcal{H}$ can be properly learned. That is, within the ZFC axioms it can be neither proven nor disproven that $\mathcal{H}$ is properly learnable. This result builds upon the breakthrough work of Ben-David et al. (2019), which established that the learnability of certain EMX (Estimating the Maximum) learning problems can be undecidable. Notably, Hanneke and Yang (2023) established an equivalence between certain EMX learning problems and bandit problems in order to establish that bandit learnability can likewise be undecidable. A related line of work, also inspired by Ben-David et al. (2019), investigates the *algorithmic decidability* of learning, i.e., examining whether problems can be learned using learners which are computable, rather than merely abstract mathematical functions (Agarwal et al., 2020). Recent developments in this area have established that there exist VC classes which cannot be learned by any computable learner (Sterkenburg, 2022), and that learnability via (improper) computable learners is instead characterized by the *effective VC dimension*, which roughly measures the smallest cardinality $k$ for which one can always compute a behavior on any $k + 1$ distinct unlabeled points which $\mathcal{H}$ cannot ex-

press (Delle Rose et al., 2023). Notably, this characterization holds for binary classification over the domain of the natural numbers; for binary classification over more general computable metric spaces, see Ackerman et al. (2022). Further work includes that of Hasrati and Ben-David (2023) on computable online learning, Gourdeau et al. (2024) on the computability of robust PAC learning, and Caro (2023), which studies the computability of learning when learners are equipped with a restricted form of black-box access to the underlying hypothesis class $\mathcal{H}$.

## 2. Preliminaries

### 2.1. Notation

For a set $Z$, we let $Z^*$ denote the collection of all finite sequences in $Z$, i.e., $Z^* = \bigcup_{i=1}^{\infty} Z^i$. When $P$ is a statement, we let $[P]$ denote the Iverson bracket of $P$, as in

$$[P] = \begin{cases} 1 & P \text{ is true,} \\ 0 & P \text{ is false.} \end{cases}$$

For a natural number $n \in \mathbb{N}$, $[n]$ denotes the set $\{1, 2, \ldots, n\}$. When $M$ is a measurable space, we let $\Delta(M)$ denote the collection of all probability measures over $M$. Finite sets $M_{\text{fin}}$ are thought of as measurable spaces by endowing them with the discrete $\sigma$-algebra by default. $\text{Unif}(M_{\text{fin}})$ denotes the uniform distribution over $M_{\text{fin}}$.

### 2.2. Learning Theory

Throughout, we use $\mathcal{X}$ to denote the **domain** in which unlabeled datapoints reside, and $\mathcal{Y}$ to denote the **label set**. A labeled datapoint is a pair $(x, y) \in \mathcal{X} \times \mathcal{Y}$. We may refer to both labeled and unlabeled datapoints merely as *datapoints* when clear from context. A **training set** is a sequence of labeled datapoints $S \in (\mathcal{X} \times \mathcal{Y})^*$. A function $f : \mathcal{X} \to \mathcal{Y}$ is a **predictor** or **hypothesis**, and a **hypothesis class** is a collection of such functions $\mathcal{H} \subseteq \mathcal{Y}^{\mathcal{X}}$. We will refer to a convex combination of hypotheses in $\mathcal{H}$ as a **randomized hypothesis** in $\mathcal{H}$. Learning makes use of a **loss function** $\ell : \mathcal{Y} \times \mathcal{Y} \to \mathbb{R}_{\geq 0}$ quantifying the quality of a predicted label $\hat{y}$ relative to the true label $y$. We will typically employ the 0-1 loss function $\ell_{0-1}(y, \hat{y}) = [y \neq \hat{y}]$ used in **multiclass classification**, but we will occasionally permit $\ell$ to be any bounded metric.

The underlying data-generating process is modeled using a probability distribution $\mathcal{D}$ over the domain $\mathcal{X}$, along with a choice of true labeling function $h^* \in \mathcal{H}$ which assigns labels to datapoints drawn from $\mathcal{D}$. For such a pair $(\mathcal{D}, h^*)$, we let $\mathcal{D}_{h^*}$ denote the distribution over $\mathcal{X} \times \mathcal{Y}$ which draws unlabeled data from $\mathcal{D}$ and labels it using $h^*$. That is, $\mathbb{P}_{\mathcal{D}_h^*}(A) = \mathbb{P}_{x \sim \mathcal{D}}\big((x, h^*(x)) \in A\big)$. We will often refer to such an $h^*$ as the "ground truth" hypothesis. Notably, we focus on the case of **realizable** learning throughout the paper, in which the data is labeled by a hypothesis in $\mathcal{H}$. For a given predictor $f : \mathcal{Y} \to \mathcal{X}$, its **true error** or simply *error* incurred with respect to the previous data-generating process is defined as the average loss it incurs on a fresh datapoint drawn from $\mathcal{D}$ and labeled by $h^*$, i.e.,

$$L_{\mathcal{D}_{h^*}}(f) = \mathop{\mathbb{E}}_{x \sim \mathcal{D}} \Big[\ell(f(x), h^*(x))\Big].$$

Similarly, the **empirical risk** incurred by $f$ on a training set $S = \big((x_1, y_1), \ldots, (x_n, y_n)\big)$ is the average loss it experiences on the datapoints in $S$,

$$L_S(f) = \frac{1}{n} \sum_{i=1}^{n} \ell(f(x_i), y_i).$$

A **learner** $A$ is a (possibly randomized) map from training sets to predictors, as in $A : (\mathcal{X} \times \mathcal{Y})^* \to \mathcal{Y}^{\mathcal{X}}$. Note that $A$ is permitted to emit predictors which are not elements of the underlying hypothesis class $\mathcal{H}$. A learner which happens to always output hypotheses in $\mathcal{H}$ is referred to as **proper**, while those which do not are **improper**.

A successful learner is one which attains vanishingly small error when trained on increasingly large datasets, as formalized by Valiant's celebrated Probably Approximately Correct (PAC) learning model (Valiant, 1984).

**Definition 1** *A hypothesis class $\mathcal{H} \subseteq \mathcal{Y}^{\mathcal{X}}$ is **PAC learnable** if there exists a learner $A$ and **sample function** $m \colon (0, 1)^2 \to \mathbb{N}$ with the following property: For any $\epsilon, \delta \in (0, 1)$, any distribution $\mathcal{D}$ over $\mathcal{X}$, and any true labeling function $h^* \in \mathcal{H}$, when $A$ is trained on a dataset $S$ of points drawn i.i.d. from $\mathcal{D}$ and labeled by $h^*$ with $|S| \geq m(\epsilon, \delta)$, then*

$$L_{\mathcal{D}_{h^*}}\big(A(S)\big) \leq \epsilon$$

*with probability at least $1 - \delta$ over the random choice of $S$ and any internal randomness in $A$.*

**Definition 2** *The **sample complexity** of a PAC learner $A$ for a class $\mathcal{H}$, denoted $m_A$, is its pointwise minimal sample function. That is, $m_A(\epsilon, \delta)$ is defined to be the smallest $n \in \mathbb{N}$ such that, for any distribution $\mathcal{D}$ and true labeling function $h^*$,*

$$L_{\mathcal{D}_{h^*}}\big(A(S)\big) \leq \epsilon$$

*with probability at least $1 - \delta$ over the choice of $|S| \geq n$ and any randomness internal to $A$.*

**Definition 3** *The **sample complexity** of a hypothesis class $\mathcal{H} \subseteq \mathcal{Y}^{\mathcal{X}}$, denoted $m_{\mathcal{H}}$, is the pointwise minimal sample complexity enjoyed by any of its learners, i.e.,*

$$m_{\mathcal{H}}(\epsilon, \delta) = \min_A m_A(\epsilon, \delta),$$

*where $A$ ranges over all learners for $\mathcal{H}$.*

In addition to the PAC model, which emphasizes high-probability guarantees, we will often judge learners' performance based upon their expected error guarantees.

**Definition 4** *Let $A$ be a learner for a hypothesis class $\mathcal{H}$. The sample complexity of $A$ in the expected error model, denoted $m_{\mathrm{Exp}, A}$, is defined by*

$$m_{\mathrm{Exp}, A}(\epsilon) = \min \left\{ m \in \mathbb{N} : \underset{S \sim \mathcal{D}_{h^*}^{m'}}{\mathbb{E}} L_{\mathcal{D}_{h^*}}\big(A(S)\big) \leq \epsilon \text{ for all } m' \geq m, \mathcal{D} \in \Delta(\mathcal{X}), h^* \in \mathcal{H} \right\}.$$

*The sample complexity of $\mathcal{H}$ in the expected error model, denoted $m_{\mathrm{Exp}, \mathcal{H}}$, is the minimal sample complexity attained by any of its learners, i.e., $m_{\mathrm{Exp}, \mathcal{H}}(\epsilon) = \min_A m_{\mathrm{Exp}, A}(\epsilon)$.*

Our results in Section 3 refer to randomized proper learners which are governed by a generalized form of regularization which we term *distributional regularization*. For reference, we recall regularization in its classic form.

**Definition 5** *A **regularizer** for a hypothesis class $\mathcal{H}$ is a function $\psi\colon \mathcal{H} \to \mathbb{R}_{\geq 0}$. A learner $A$ for $\mathcal{H}$ is a **structural risk minimizer** (SRM) if there exists a regularizer $\psi$ for $\mathcal{H}$ such that for all training samples $S$,*

$$A(S) \in \arg\min_{\mathcal{H}} L_S(h) + \psi(h).$$

In the realizable setting, SRM learners are sometimes defined as those which minimize the regularization value $\psi(h)$ subject to a *hard* constraint on attaining zero training error (Asilis et al., 2024a). This perspective is essentially equivalent to Definition 5, as one can normalize $\psi$ to have output strictly less than $\frac{1}{|S|}$ for the case of classification. (Note that this normalization depends upon $|S|$, however.)

## 3. Proper Learning Through Distributional Regularization

We begin by establishing a sufficient condition for proper learnability, based upon knowledge of the marginal distribution $\mathcal{D}$ over unlabeled datapoints. First, we observe that when the learner is granted full knowledge of $\mathcal{D}$, then a hypothesis class $\mathcal{H}$ can always be learned by a proper learner with optimal sample complexity, as measured in the expected error model. Next, we shed light on the particular algorithm form of such learners by demonstrating that one such learner always exists which is governed by *distributional regularization* – a form of regularization which assigns a complexity score to randomized hypotheses in $\mathcal{H}$ (i.e., to convex combinations of hypotheses in $\mathcal{H}$). Our results hold for domains $\mathcal{X}$ and label sets $\mathcal{Y}$ of arbitrary finite size, with no dependence upon their (finite) cardinalities. We conjecture that our results hold for more general choices of $\mathcal{X}$ and $\mathcal{Y}$, perhaps via topological arguments. Throughout the section, we remain in the setting of realizable learning.

First, let us introduce *distribution-fixed* learning. In short, it is a modification of PAC learning in which the learner is given complete information regarding the marginal distribution $\mathcal{D}$ over unlabeled data. Notably, however, $\mathcal{D}$ is permitted to be entirely arbitrary, and the learner will be judged on a worst-case basis over all possible choices of $\mathcal{D}$, as we now describe.

**Definition 6** *A **distribution-fixed learner** is a function $A\colon \Delta(\mathcal{X}) \times (\mathcal{X} \times \mathcal{Y})^* \to \mathcal{Y}^{\mathcal{X}}$, that is, a function which receives both a training sample and a probability distribution over $\mathcal{X}$, and emits a predictor.*

**Definition 7** *A hypothesis class $\mathcal{H} \subseteq \mathcal{Y}^{\mathcal{X}}$ is **distribution-fixed PAC learnable** if there exists a distribution-fixed learner $A$ and function $m\colon (0,1)^2 \to \mathbb{N}$ with the following property: For any $\epsilon, \delta \in (0,1)$, any distribution $\mathcal{D}$ over $\mathcal{X}$, and any true labeling function $h^* \in \mathcal{H}$, when $S$ is a training set of at least $m(\epsilon, \delta)$ many points drawn i.i.d. from $\mathcal{D}$ and labeled by $h^*$, then*

$$L_{\mathcal{D}_{h^*}}\big(A(\mathcal{D}, S)\big) \leq \epsilon$$

*with probability at least $1 - \delta$ over the random choice of $S$ and any internal randomness in $A$.*

As in classical PAC learning, the minimal function $m$ satisfying Definition 7 is the *sample complexity* of the learner $A$, and the minimal such function across all learners for $\mathcal{H}$ is the sample complexity of $\mathcal{H}$. (Note too that the expected error model of Definition 4 can likewise be made distribution-fixed in the natural way.)

**Remark 8** *The distribution-fixed model reflects the setting in which the marginal distribution over unlabeled data, $\mathcal{D}$, is fully known to the learner at training time, yet the learner is judged on its worst-case performance across any choice of $\mathcal{D}$ (and true labeling function $h^* \in \mathcal{H}$). Other models, including the seminal work of Benedek and Itai (1991), focus on the case in which the marginal $\mathcal{D}$ does* not *vary (and only the true labeling function can vary). In this setting, the learner is endowed with complete information of $\mathcal{D}$ "by default", i.e., because its performance is only examined on distributions which share this marginal. In short, Benedek and Itai (1991) adopt an instance-optimal perspective on learning under a particular marginal distribution. The version we study can be thought of as intermediate between the classical PAC model and that of Benedek and Itai. Informally, we maintain a worst-case perspective but equip learners with a complete understanding of the unlabeled data.*

We now present a somewhat striking result: for any bounded loss function, the distribution-fixed and classical PAC models are equivalent at the level of learnability, and furthermore have sample complexities which differ by at most a logarithmic factor.

**Theorem 9 (Equivalence between distribution-fixed and classical PAC models)** *Let $\mathcal{X}$ be an arbitrary domain, $\mathcal{Y}$ an arbitrary label set, and $\mathcal{H} \subseteq \mathcal{Y}^{\mathcal{X}}$ a hypothesis class. Employ a loss function $\ell \colon \mathcal{Y} \times \mathcal{Y} \to \mathbb{R}_{\geq 0}$ which is bounded in $[0,1]$. Let $m_{\mathcal{H}}$ denote the sample complexity of $\mathcal{H}$ in the classic PAC model and $m_{\mathcal{H}}^{\mathrm{DF}}$ its sample complexity in the distribution-fixed PAC model. Then,*

$$m_{\mathcal{H}}^{\mathrm{DF}}(\epsilon, \delta) \leq m_{\mathcal{H}}(\epsilon, \delta) \leq O\left(m_{\mathcal{H}}^{\mathrm{DF}}\left(\frac{\epsilon}{11}, \frac{\epsilon}{11}\right) \cdot \log(1/\delta)\right).$$

*Furthermore, if $m_{\mathrm{Exp}, \mathcal{H}}(\epsilon)$ and $m_{\mathrm{Exp}, \mathcal{H}}^{\mathrm{DF}}(\epsilon)$ denote the sample complexities of learning $\mathcal{H}$ to expected error $\leq \epsilon$ in the classic and distribution-fixed models, respectively, then*

$$m_{\mathrm{Exp}, \mathcal{H}}^{\mathrm{DF}}(\epsilon) \leq m_{\mathrm{Exp}, \mathcal{H}}(\epsilon) \leq m_{\mathrm{Exp}, \mathcal{H}}^{\mathrm{DF}}(\epsilon/e),$$

*where $e \approx 2.718$ is Euler's number.*

**Proof sketch** We defer the proof of the second set of inequalities to Appendix A.1. For the first set of inequalities, begin by noting that $m_{\mathcal{H}}^{\mathrm{DF}}(\epsilon, \delta) \leq m_{\mathcal{H}}(\epsilon, \delta)$ is immediate; a distribution-fixed learner can elect to ignore the information of the marginal $\mathcal{D}$. Then, assuming the second set of inequalities, we have:

$$\begin{aligned} m_{\mathcal{H}}(\epsilon, \delta) &\leq O\big(m_{\mathrm{Exp}, \mathcal{H}}(\epsilon/2) \log(1/\delta)\big) \\ &\leq O\left(m_{\mathrm{Exp}, \mathcal{H}}^{\mathrm{DF}}\left(\frac{\epsilon}{2e}\right) \log(1/\delta)\right) \\ &\leq O\left(m_{\mathcal{H}}^{\mathrm{DF}}\left(\frac{\epsilon}{4e}, \frac{\epsilon}{4e}\right) \log(1/\delta)\right). \end{aligned}$$

The first inequality makes use of a standard repetition argument; a learner incurring expected error at most $\epsilon/2$ can be repeatedly trained on separate datasets and tested on a validation set in order

to attain a high-probability guarantee. The second inequality invokes the claim whose proof we deferred to Appendix A.1. The third inequality follows immediately from the fact that the loss function is bounded above by 1. Conclude by noting that $4e \approx 10.87 < 11$. ∎

We now observe a simple equivalence between proper and improper learnability in the distribution-fixed model, for learning with any metric loss function.[2] In particular, an arbitrary (possibly improper) distribution-fixed learner $\mathcal{A}$ can be "properized" by replacing its output $\mathcal{A}(\mathcal{D}, S)$ with the nearest hypothesis in $\mathcal{H}$, as measured by the distance function $\text{dist}_\mathcal{D}(f, g) = \mathbb{E}_{x \sim \mathcal{D}} \ell\big(f(x), g(x)\big)$. Notice, however, that this settles the question of proper learnability in the distribution-fixed model, but says little about the *algorithmic form* of (optimal) proper learners. In Theorem 14, in contrast, we shed light upon such learners as following the principle of regularization, in a generalized form.

**Observation 10** *Let $\mathcal{X}$ be an arbitrary domain, $\mathcal{Y}$ an arbitrary label set, and $\mathcal{H} \subseteq \mathcal{Y}^\mathcal{X}$ a hypothesis class. Employ a metric loss function $\ell \colon \mathcal{Y} \times \mathcal{Y} \to \mathbb{R}_{\geq 0}$. Then $\mathcal{H}$ has a proper distribution-fixed learner which is optimal, as measured by its expected error.*

**Proof** Let $\mathcal{A} \colon \Delta(\mathcal{X}) \times (\mathcal{X} \times \mathcal{Y})^* \to \mathcal{Y}^\mathcal{X}$ be an optimal distribution-fixed learner for $\mathcal{H}$, in the expected error regime. We will exhibit a proper distribution-fixed learner $\mathcal{B}$ which attains equal performance to $\mathcal{A}$, up to a factor of 2. To this end, let $\mathcal{D} \in \Delta(\mathcal{X})$ be an arbitrary probability measure on $\mathcal{X}$ and $S$ a training set. Let $\text{dist}_\mathcal{D} \colon \mathcal{Y}^\mathcal{X} \times \mathcal{Y}^\mathcal{X} \to \mathbb{R}_{\geq 0}$ be the distance measure defined as $\text{dist}_\mathcal{D}(f, g) = \mathbb{E}_{x \sim \mathcal{D}} \ell\big(f(x), g(x)\big)$.

Then we define $\mathcal{B}$ to emit the following hypothesis on input pair $(\mathcal{D}, S)$:

$$\mathcal{B}(\mathcal{D}, S) = \arg\min_{\mathcal{H}} \Big[ \text{dist}_\mathcal{D}\big(h, \mathcal{A}(\mathcal{D}, S)\big) \Big].$$

To see that $\mathcal{B}$ at most doubles the expected error of $\mathcal{A}$ on any realizable distribution, fix one such distribution, as defined by a marginal $\mathcal{D} \in \Delta(\mathcal{X})$ and ground truth hypothesis $h^* \in \mathcal{H}$. Then we have,

$$\begin{aligned}
L_{\mathcal{D}_{h^*}}\big(\mathcal{B}(\mathcal{D}, S)\big) &= \mathbb{E}_{x \sim \mathcal{D}} \ell\big(\mathcal{B}(\mathcal{D}, S), h^*(x)\big) \\
&\leq \mathbb{E}_{x \sim \mathcal{D}} \ell\big(\mathcal{B}(\mathcal{D}, S), \mathcal{A}(\mathcal{D}, S)\big) + \mathbb{E}_{x \sim \mathcal{D}} \ell\big(\mathcal{A}(\mathcal{D}, S), h^*(x)\big) \\
&\leq \mathbb{E}_{x \sim \mathcal{D}} \ell\big(\mathcal{A}(\mathcal{D}, S), h^*(x)\big) + \mathbb{E}_{x \sim \mathcal{D}} \ell\big(\mathcal{A}(\mathcal{D}, S), h^*(x)\big) \\
&= 2 \cdot L_{\mathcal{D}_{h^*}}\big(\mathcal{A}(\mathcal{D}, S)\big).
\end{aligned}$$

The first inequality is an application of the triangle inequality for $\ell$, and the second inequality follows from the definition of $\mathcal{B}$. ∎

We now define *distributional regularization*, a relaxation of classical regularization which assigns values to randomized hypotheses in $\mathcal{H}$ (i.e., to probability distributions over $\mathcal{H}$).

**Definition 11** *A **distributional regularizer** is a function $\psi \colon \Delta(\mathcal{H}) \to \mathbb{R}_{\geq 0}$. A (randomized) learner $A$ for $\mathcal{H}$ is a **distributional structural risk minimizer (SRM)** if there exists a distributional regularizer $\psi$ such that for all training samples $S$,*

$$A(S) \in \arg\min_{\substack{P \in \Delta(\mathcal{H}), \\ \mathbb{E}_{h \sim P} L_S(h) = 0}} \psi(P).$$

---

2. We thank Tosca Lechner for pointing us to this elegant observation.

That is, a distributional SRM learner is one which outputs a randomized hypothesis minimizing the regularization value, subject to perfect performance on the training set. When there are ties in the regularization value $\psi(.)$, we evaluate the learner's performance with respect to worst-case tie-breaking among randomized hypotheses. An important lemma is that Bayesian learners can be witnessed as distributional SRMs.

**Lemma 12** *Let $\mathcal{X}$ be a finite domain, $\mathcal{Y}$ a finite label space, and $\mathcal{H} \subseteq \mathcal{Y}^{\mathcal{X}}$ a hypothesis class. Let $Q \in \Delta(\mathcal{H})$ be a distribution over $\mathcal{H}$, and $A$ be a Bayesian learner with respect to Q. That is, upon receiving a training sample $S$, $A$ emits $Q|S \in \Delta(\mathcal{H})$, the restriction of $D$ to those $h \in \mathcal{H}$ with $L_S(h) = 0$. Then $A$ is a distributional SRM learner.*

**Proof** Given $Q$, let $\psi$ be the distributional regularizer which computes the relative entropy of a distribution $P \in \Delta(\mathcal{H})$ with respect to $Q$. That is,

$$\psi(P) = D_{KL}(P|Q)$$
$$= \sum_{h \in \mathcal{H}} P(h) \log\left(\frac{P(h)}{Q(h)}\right).$$

Then an SRM learner induced by $\psi$ is tasked with outputing a distribution $P$ which minimizes empirical error (i.e., is supported on $L_S^{-1}(0)$) while minimizing relative entropy to $Q$. By a standard result, the distribution supported on $L_S^{-1}(0)$ with minimal relative entropy to $Q$ is precisely $Q|S$, the restriction of $Q$ to $L_S^{-1}(0)$. (See, e.g., Asilis et al. (2024b, Lemma 55).) This completes the argument. ∎

We will also make use of the fact that Bayesian learners, as described in Lemma 12, are closed under convex combinations.

**Lemma 13** *Let $\mathcal{X}$ be a finite domain, $\mathcal{Y}$ a finite label space, and $\mathcal{H} \subseteq \mathcal{Y}^{\mathcal{X}}$ a hypothesis class. Let $A_1, \ldots, A_n$ be a collection of randomized learners for $\mathcal{H}$ which are Bayesian with respect to priors $Q_1, \ldots, Q_n$. Then any convex combination $p_1 A_1 + \ldots + p_n A_n$ is itself a Bayesian learner with respect to the prior $p_1 Q_1 + \ldots + p_n Q_n$.*

**Proof** Fix a probability distribution $(p_1, \ldots, p_n)$ and a training set $S$. Let $\mathcal{A} = p_1 A_1 + \ldots + p_n A_n$, and let $\mathcal{A}'$ be the Bayesian learner corresponding to the prior $p_1 Q_1 + \ldots + p_n Q_n$. For any $h \in \mathcal{H}$, we have:

$$\mathcal{A}(S)(h) = p_1 A_1(S)(h) + \ldots + p_n A_n(S)(h)$$
$$= p_1 \cdot Q_1(h \mid L_S^{-1}(0)) + \ldots + p_n \cdot Q_n(h \mid L_S^{-1}(0))$$
$$= (p_1 Q_1 + \ldots + p_n Q_n)(h \mid L_S^{-1}(0))$$
$$= \mathcal{A}'(S)(h).$$

∎

We now demonstrate the primary result of the section: all finite learning problems with bounded loss functions can be learned by an *optimal* randomized proper learner, following the principle of distributional regularization.

**Theorem 14 (Distributional regularization)** *Let $\mathcal{X}$ be a finite domain, $\mathcal{Y}$ a finite label set, and $\mathcal{H} \subseteq \mathcal{Y}^{\mathcal{X}}$ a hypothesis class. Let $\ell \colon \mathcal{Y} \times \mathcal{Y} \to \mathbb{R}_{\geq 0}$ be a metric loss function bounded in $[0, 1]$. Then in the distribution-fixed model, $\mathcal{H}$ has a randomized proper learner which attains optimal expected error, up to a factor of 2. Furthermore, this learner can be witnessed as a distributional SRM.*

**Proof** Fix a sample size $n \in \mathbb{N}$. We will describe the action of a distributional SRM learner $\mathcal{A}$ which attains optimal expected error on samples of size $n$, up to factor of 2. Further fix a distribution $\mathcal{D}$ over $\mathcal{X}$, of which $\mathcal{A}$ is aware, as we are in the distribution-fixed model.

Consider the zero-sum game $\mathcal{G}$ in which the column player selects a function $h^* \in \mathcal{H}$, thought of as the ground truth labeling function, and the row player responds with a learner $A : (\mathcal{X} \times \mathcal{Y})^n \to \mathcal{Y}^{\mathcal{X}}$, defined only on samples of size $n$. For a given pair of actions $(h^*, A)$, the row player incurs a loss of

$$\epsilon(h^*, A) = \mathop{\mathbb{E}}_{S \sim \mathcal{D}_{h^*}^n} \left[ L_{\mathcal{D}_{h^*}} A(S) \right]$$
$$= \mathop{\mathbb{E}}_{S \sim \mathcal{D}_{h^*}^n} \mathop{\mathbb{E}}_{x \sim \mathcal{D}} \left[ \ell\big( h^*(x), A(S)(x) \big) \right],$$

i.e., the expected error incurred by $A$ when trained on samples of size $n$. As $\mathcal{G}$ is zero-sum, the column player is rewarded with a value of $\epsilon(h^*, A)$. Note too that $\mathcal{G}$ is a finite game; there are only finitely many hypotheses in $\mathcal{H} \subseteq \mathcal{Y}^{\mathcal{X}}$ — owing to finiteness of both $\mathcal{X}$ and $\mathcal{Y}$ — and likewise finitely many learners $(\mathcal{X} \times \mathcal{Y})^n \to \mathcal{Y}^{\mathcal{X}}$.

> **Lemma 15** *For any mixed strategy $\Lambda = \{\lambda_h\}_{h \in \mathcal{H}}$ of the column player in game $\mathcal{G}$, the row player can respond with a randomized proper learner $A_{prop}$ which attains optimal payoff, up to a factor of 2. The learner $A_{prop}$ is Bayesian and employs $\Lambda$ as its prior.*
>
> **Proof** Fix the mixed strategy $\Lambda = \{\lambda_h\}_{h \in \mathcal{H}}$, denoting a prior probability distribution over the ground truth labeling function $h^* \in \mathcal{H}$. For a training set $S$ and test point $x$, $A_{\text{prop}}$ predicts a label for $x$ by drawing from the posterior distribution over labels at $x$. (Equivalently, $A_{\text{prop}}$ emits an entire function $h \in \mathcal{H}$ which is drawn from the posterior distribution of $\Lambda$ conditioned upon $S$.)
>
> We now argue that the expected error incurred by $A_{\text{prop}}$ is optimal, up to a factor of 2. To this end, fix a test point $x \in \mathcal{X}$ and let $P = \{p_y\}_{y \in \mathcal{Y}}$ denote the posterior distribution over the true label at $x$ (upon conditioning $\Lambda$ by $S$). By linearity of expectation, the optimal prediction at $x$ can be assumed to be deterministic, i.e., predicting a fixed label $\hat{y} \in \mathcal{Y}$. Thus, the expected error incurred by the optimal learner at test point $x$ is $\mathbb{E}_{y \sim P} \ell(y, \hat{y})$. The error incurred by $A_{\text{prop}}$, in contrast, is $\mathbb{E}_{y, y' \sim P} \ell(y, y')$. With an application of the triangle inequality, we have:
>
> $$\mathop{\mathbb{E}}_{y, y' \sim P} \ell(y, y') \leq \mathop{\mathbb{E}}_{y, y' \sim P} \ell(y, \hat{y}) + \ell(\hat{y}, y')$$
> $$= 2 \mathop{\mathbb{E}}_{y \sim P} \ell(y, \hat{y}).$$
>
> Thus $A_{\text{prop}}$ indeed incurs expected error at most twice that of the optimum. ∎

Now consider the game $\mathcal{G}_{\text{prop}}$ which is identical to $\mathcal{G}$, save for the fact that the row player is obligated to select a Bayesian learner. Then $\mathcal{G}_{\text{prop}}$ is a compact zero-sum game by the supposition that $\mathcal{X}$ and $\mathcal{Y}$ are finite, meaning it enjoys the minimax theorem. There thus exists an optimal mixed strategy over Bayesian learners which attains the value $\epsilon^*$ of $\mathcal{G}_{\text{prop}}$. By Lemma 13, the mixed strategy reduces to a pure strategy; that is, there exists a single Bayesian learner attaining the value $\epsilon^*$. Furthermore, by Lemma 15, $\epsilon^*$ is within a factor 2 of the value of $\mathcal{G}$, i.e., of the best performance which can be attained by *any* learner on samples of size $n$. Conclude by applying Lemma 12 to see that Bayesian learners can be witnessed as distributional SRM learners, as desired. ∎

**Remark 16** *We note that our proof of Theorem 14 bears similarities to the proof of Darnstädt and Simon (2011, Lemma 5), which also models learning with respect to a fixed marginal distribution as a zero-sum game. Though the goals of their paper and this particular lemma are substantially different from ours, their analysis of the learner's best response problem is conceptually similar. One key difference is that their proof exploits structure particular to binary classification, which is their focus. Another difference is that they restrict attention to proper learning out of the gate, as this is without loss for binary classification, whereas we allow improper learning and conclude that properness is without much loss for more general problem classes. Finally, we aggregate the (Bayesian, and proper) best responses of the learner into a near-optimal learner of the same desired form, departing from the concerns and technical approach of their paper.*

Let us briefly remark upon two structural features of Theorem 14's proof. First, the fact that a learner $\mathcal{A}$ has completely flexible control over its strategy in the zero-sum game $\mathcal{G}$ relies crucially upon the fact that $\mathcal{A}$ is a distribution-fixed learner. Otherwise, $\mathcal{A}$'s actions would be coupled across all possible marginals $\mathcal{D}$ used in the definition of $\mathcal{G}$. Second, it is tempting to generalize Theorem 14's proof to more general settings, including spaces $\mathcal{X}$ and $\mathcal{Y}$ which may be compact, convex, etc. This is a interesting direction which we leave open to future work. We mention only that in our attempts to do so, we were unable to find natural choices of structure on $\mathcal{X}$, $\mathcal{Y}$, $\mathcal{H}$, and $\ell$ which could simultaneously satisfy all properties required in defining the game and invoking the minimax theorem. Another interesting question is whether Theorem 14 can also be established for the high-probability regime of learning.

**Weakening of the distribution-fixed assumption.** It is natural to ask whether the conclusion of Theorem 14 can be achieved by assuming a "softer" form of distribution-fixed learning. For instance, perhaps one can remain in the classic PAC model yet assume that $\mathcal{H}$ is sufficiently simple such that a learner $A$ can use the unlabeled data in $S$ in order to learn the marginal distribution $\mathcal{D}$ over $\mathcal{X}$ in some "class-conditional" sense. (E.g., to learn $\mathcal{D}$ sufficiently well so as to estimate $L_{\mathcal{D}_h}(h')$ for all pairs $(h, h') \in \mathcal{H}^2$.) This line of inquiry is studied by Hopkins et al. (2023) for binary classification in the *distribution-family* model, in which the marginal distribution $\mathcal{D}$ over $\mathcal{X}$ is restricted to a certain collection of distributions at the outset. They introduce precisely such "class-conditional" notions of learning the marginal $\mathcal{D}$, and provide distinct necessary and sufficient conditions for PAC learnability based upon learnability of $\mathcal{D}$.

Notably, however, the difficulty in Hopkins et al. (2023) arises from studying the distribution-family model, and the complexity which it can endow binary classification. Further, Hopkins et al. (2023) do not emphasize the particular algorithmic or structural form of the learner, as we do. In

PAC learning with respect to *all* realizable distributions — as we study — binary classification problems are also well-known to be learnable by proper learners whenever learning is possible.

It may be natural to ask, then, whether the techniques of Hopkins et al. (2023) are applicable for classical PAC learning beyond the binary setting. To this, we present a negative result in Section 4, by demonstrating that proper learnability is not a *monotone* property. That is, there exist hypothesis classes $\mathcal{H}_0 \subsetneq \mathcal{H}_1 \subsetneq \mathcal{H}_2$ in multiclass classification such that only $\mathcal{H}_0$ and $\mathcal{H}_2$ are properly learnable. All notions of class-conditional learnability introduced by Hopkins et al. (2023), however, are monotone (e.g., Weak TV-learning, Strong TV-learning, Exact TV-learning). As such, any natural weakening of the distribution-fixed assumption — reflecting the ability to learn $\mathcal{D}$ in a "class-conditional" way — is unlikely to characterize proper learnability.

**Proper learnability and SRM in broader context.** A central point of Theorem 14 is that the proper learner is *optimal* in terms of its expected error, up to a constant factor of 2. In particular, even for *finite* problems in classical PAC learning (i.e., finite $\mathcal{X}$ and $\mathcal{Y}$), there are known to be problems exhibiting arbitrarily large gaps in sample complexity between proper and improper learners. (See Daniely and Shalev-Shwartz (2014, Theorem 1), along with the compactness result of Asilis et al. (2024c) which equates the sample complexity of a learner to its worst-case over finite subproblems.) As such, Theorem 14 would not hold when stated in the classic PAC model. Furthermore, Theorem 14 establishes an equivalence between proper learnability and learnability by (distributional) SRM in the distribution-fixed model. It is natural to ask whether such an equivalence might hold more generally, such as in the classic PAC model. We conjecture that it may indeed be so, at least for the case of multiclass classification.

**Conjecture 17** *Let $\mathcal{X}$ be an arbitrary domain, $\mathcal{Y}$ an arbitrary label space, and employ the 0-1 loss function $\ell_{0-1}$. Then for any $\mathcal{H} \subseteq \mathcal{Y}^{\mathcal{X}}$, $\mathcal{H}$ is properly learnable if and only if $\mathcal{H}$ can be learned by an SRM learner. (Perhaps the same can be said if $\ell$ is any bounded metric loss function.)*

Let us present another conjecture, which — along with the previous one — would imply that *all* classification problems $\mathcal{H}$ can be learned by SRM, possibly on a superset of $\mathcal{H}$.

**Conjecture 18** *Let $\mathcal{H} \subseteq \mathcal{Y}^{\mathcal{X}}$ be a multiclass classification problem (i.e., employ $\ell_{0-1}$). Then $\mathcal{H}$ is learnable if and only if there exists an $\mathcal{H}' \supseteq \mathcal{H}$ such that $\mathcal{H}'$ is properly learnable.*

## 4. Obstructions to Characterizing Proper Learnability

We now direct our attention from the distribution-fixed PAC to the classical PAC model, and ask: Under what conditions is a learning problem *properly* learnable? Perhaps the most natural approach is to search for a combinatorial characterization of proper learnability, by analogy with existing characterizations of (improper) PAC learnability. We demonstrate several obstructions to any such approach. Together, they imply that proper learnability cannot be characterized by any *property of finite character*, as described in Ben-David et al. (2019). Our results also imply, more generally, that proper learnability cannot be characterized by any condition which is monotone, or which considers only "finite projections" of the hypothesis class. Throughout the section, we remain in the setting of multiclass classification in the (realizable) PAC model. Along with binary classification, this forms perhaps the most fundamental setting of supervised learning.

We now demonstrate a central technical result of the section: the pathological learning problem of *EMX learning* — which was not originally phrased as a supervised learning problem (Ben-David

et al., 2019) — can in fact be witnessed as an instance of proper multiclass learning. We first recall the standard definition of EMX learning, and the result for which it was designed: EMX learnability can be logically undecidable.

**Definition 19** *Let $\mathcal{F}$ be a set. The EMX learning problem on $\mathcal{F}$ is defined as follows: An adversary selects a probability distribution $P$ on $\mathcal{F}$ which is supported on finitely many points, the learner receives a sample $S$ of points drawn i.i.d. from $P$, and it must emit a finite subset of $\mathcal{F}$ with large $P$-measure.*

The learning problem associated to $\mathcal{F}$ is said to be EMX-learnable if there exists a learner which outputs sets of measure arbitrarily close to 1 as $|S| \to \infty$, with high probability over $S$ (and uniformly in the adversary's choice of $P$). We will often abbreviate this by stating that $\mathcal{F}$ itself is, or is not, EMX-learnable. A breakthrough result of Ben-David et al. (2019) demonstrated that EMX learning can be undecidable, depending upon the cardinality of the underlying set $\mathcal{X}$.

**Theorem 20 (Ben-David et al. (2019))** *The EMX-learnability of $\mathbb{R}$ is undecidable, i.e., logically independent of the ZFC axioms. More generally, a set $\mathcal{F}$ is EMX-learnable if and only if $|\mathcal{F}| < \aleph_\omega$.*

We now demonstrate that EMX learning can be witnessed within multiclass classification. Our proof employs techniques developed by Daniely et al. (2015) and Daniely and Shalev-Shwartz (2014) in their design of the *first Cantor class*.

**Proposition 21** *Let $\mathcal{F}$ be any set. There exists a multiclass classification problem $\mathcal{H} \subseteq \mathcal{Y}^{\mathcal{X}}$ such that $\mathcal{H}$ is properly learnable if and only if $\mathcal{F}$ is EMX-learnable.*

**Proof** Set $\mathcal{X} = \mathcal{F}$ and $\mathcal{Y} = \{\star\} \cup 2^{\mathcal{F}}$, where $2^{\mathcal{F}}$ denotes the power set of $\mathcal{F}$. For each $A \subseteq \mathcal{F}$, define $h_A : \mathcal{X} \to \mathcal{Y}$ by

$$h_A(x) = \begin{cases} A & x \in A, \\ \star & x \notin A. \end{cases}$$

Then set $\mathcal{H} = \{h_A : A \subseteq \mathcal{X}, |\mathcal{X} \setminus A| < \infty\}$. That is, each $h_A \in \mathcal{H}$ outputs the label $\star$ on a finite set of points, and the label $A$ elsewhere. Note that the label $A$, in only being output by the hypothesis $h_A$, completely reveals the identity of $h_A$ as the true labeling function. In fact, if a learner ever observes any label other than $\star$ in the training set, then it has fully identified the true labeling function. Then in order to learn $\mathcal{H}$, one need only consider learnability with respect to pairs $(D, h)$ where $D$ places full measure on $h^{-1}(\star)$. In particular, upon observing a training set $S$ with $|S| = n$, either $S$ contains a non-$\star$ label, rendering learning trivial, or $S$ only contains the $\star$ label, from which one can conclude that there is $o_n(1)$ probability of ever observing a non-$\star$ label (and thus learning reduces to correctly predicting the $\star$ labels).[3]

Now suppose that $\mathcal{H}$ is properly learnable. Then there exists a learner $\mathcal{A}$ for $\mathcal{H}$ with the following property: for any marginal distribution $D$ over $\mathcal{X}$ with finite support, when $\mathcal{A}$ observes a training sample $S = (x_i, \star)_{i \in [n]}$ with $x_i \overset{\text{i.i.d.}}{\sim} D$, it emits a hypothesis $h_A \in \mathcal{H}$ such that $\mathcal{X} \setminus A$ is finite and has large $D$-measure. By modifying $\mathcal{A}$ to receive only the data of $(x_i)_{i \in [n]}$ and to emit $\mathcal{X} \setminus A$ rather than $h_A$, we produce an EMX learner for $\mathcal{F}$. Conversely, any EMX learner $\mathcal{A}$ for $\mathcal{F}$ gives rise to a PAC learner for $\mathcal{H}$ by nearly identical reasoning, i.e., by reformatting its input and output. ∎

---

3. More precisely, for any distribution $D$ and sample size $n$, either $D$ places $O(\frac{1}{n})$ mass on non-$\star$ labels (thus learning reduces to correctly predicting the $\star$ label), or $S \sim D^n$ contains a non-$\star$ label with probability $1 - o(1)$.

**Remark 22** *In Proposition 21, it suffices to endow $\mathcal{X}$ with any $\sigma$-algebra such that points in $\mathcal{X}$ are measurable. From this, one has that for each $h_A \in \mathcal{H}$, $\mathcal{X} \setminus A$ is measurable (as it is finite) and $A$ is measurable (as it is cofinite). Then, as previously described, any marginal distribution $D$ either places a negligible amount of mass on the event $A$, or otherwise reveals the label $A$ exponentially quickly in $|S|$.*

**Theorem 23** *There exists a multiclass classification problem $\mathcal{H}$ such that it is undecidable whether $\mathcal{H}$ is properly learnable.*

**Proof** Invoke Theorem 20 with Proposition 21. ■

An immediate consequence of Theorem 23 is that proper multiclass learnability is not a *property of finite character*, in the sense of Ben-David et al. (2019). This stands in stark contrast to the existing characterization of improper learnability by the DS dimension, and to similar dimension-based results across supervised learning. We now demonstrate that proper learnability is furthermore not a *local* property. That is, there exist hypothesis classes sharing all local behaviors yet differing in their proper learnability.

**Theorem 24** *In multiclass classification, there exist a pair of hypothesis class $\mathcal{H}, \mathcal{H}' \subseteq \mathcal{Y}^{\mathcal{X}}$ such that $\mathcal{H}|_S = \mathcal{H}'|_S$ for each finite $S \subseteq \mathcal{X}$, yet $\mathcal{H}$ is properly learnable and $\mathcal{H}'$ is not.*

**Proof** Let $\mathcal{F}$ be a set of cardinality $|\mathcal{F}| \geq \aleph_\omega$. Let $\mathcal{H} \subseteq \mathcal{Y}^{\mathcal{X}}$ be the multiclass problem associated to $\mathcal{F}$, as per Proposition 21. By Theorem 20, $\mathcal{F}$ is not EMX-learnable and thus $\mathcal{H}$ is not properly learnable. Let $f^\star \colon \mathcal{X} \to \mathcal{Y}$ be the constant function which only outputs the $\star$ label. Then $\mathcal{H}' = \mathcal{H} \cup \{f^\star\}$ is certainly properly learnable, by the learner which outputs $f^\star$ when it only sees the $\star$ label in the training set, and otherwise sees a label $A \neq \star \in \mathcal{Y}$ which fully identifies the true labeling function as $h_A$. Yet for any finite $S \subseteq \mathcal{X}$, we have that $\mathcal{H}|_S = \mathcal{H}'|_S$, because the behavior $f^*|_S$ appears in $\mathcal{H}|_S$ as the restriction of (for instance) $h_{\mathcal{X} \setminus S}$ to $S$. This completes the argument. ■

**Theorem 25** *In multiclass classification, proper learnability is not a monotone property of the hypothesis class. That is, there exist hypothesis classes $\mathcal{H}_0 \subsetneq \mathcal{H}_1 \subsetneq \mathcal{H}_2$ such that only $\mathcal{H}_0$ and $\mathcal{H}_2$ are properly learnable.*

**Proof** Using the proof of Theorem 24, we have a pair of hypothesis classes $\mathcal{H} \subsetneq \mathcal{H}'$ such that $\mathcal{H}'$ is properly learnable yet $\mathcal{H}$ is not. Conclude by setting $\mathcal{H}_0$ to be any finite subset of $\mathcal{H}$, which is properly learnable as it satisfies the uniform convergence property. ■

## 5. Conclusion

We study proper learnability in supervised learning, and begin by considering the distribution-fixed model of learning, in which the learner is given the full information of the marginal distribution $\mathcal{D}$ over unlabeled data. We demonstrate an approximate equivalence between sample complexities in the distribution-fixed model and the classic PAC model, for any bounded metric loss function. This refutes the power of unlabeled data in PAC learning, i.e., for the worst-case distributions. We then establish that in the distribution-fixed model, all finite learning problems with metric losses can be

learned to optimal expected error by a proper learner. We conjecture that this result can be extended to infinite domains $\mathcal{X}$, perhaps via topological arguments.

We then demonstrate impossibility results towards characterizing proper learnability in the classic PAC model. Our results are threefold: we show that proper learnability can be logically undecidable, that it is not a monotone property, and that it is not a local property. This strongly suggests that a characterization of proper learnability will require fundamentally different techniques from the usual dimensions in learning theory. Furthermore, the non-monotonicity of proper learnability rules out many natural characterizations in terms of unsupervised learning, such as class-conditional learning of the unlabeled data distribution. Interesting open questions include studying proper learning in the *agnostic* case, establishing necessary or sufficient conditions for proper learnability in the classic PAC model, and understanding the algorithmic form of (optimal) proper learners.

## Acknowledgments

The authors thank Tosca Lechner for a helpful conversation and for suggesting the ideas behind Observation 10. Julian Asilis was supported by the Simons Foundation and by the National Science Foundation Graduate Research Fellowship Program under Grant No. DGE-1842487. This work was completed in part while Julian Asilis, Siddartha Devic, and Vatsal Sharan were visiting the Simons Institute for the Theory of Computing. Shaddin Dughmi was supported by NSF Grant CCF-2432219. Vatsal Sharan was supported by NSF CAREER Award CCF-2239265 and an Amazon Research Award. Shang-Hua Teng was supported in part by NSF Grant CCF-2308744 and the Simons Investigator Award from the Simons Foundation. Any opinions, findings, conclusions, or recommendations expressed in this material are those of the authors and do not necessarily reflect the views of any of the sponsors such as the NSF.

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

## Appendix A. Omitted proofs

### A.1. Proof of Theorem 9

Completing the proof of Theorem 9 amounts to establishing the following claim.

**Lemma 26** *Let $\mathcal{X}$ be an arbitrary domain, $\mathcal{Y}$ an arbitrary label space, and $\mathcal{H} \subseteq \mathcal{Y}^{\mathcal{X}}$ a hypothesis class. Employ a loss function $\ell : \mathcal{Y} \times \mathcal{Y} \to \mathbb{R}_{\geq 0}$ which is bounded in $[0, 1]$. Let $m_{\mathrm{Exp}, \mathcal{H}}(\epsilon)$ and $m_{\mathrm{Exp}, \mathcal{H}}^{\mathrm{DF}}(\epsilon)$ denote the sample complexity of learning $\mathcal{H}$ to expected error $\leq \epsilon$ in the classic and distribution-fixed models, respectively. Then,*

$$m_{\mathrm{Exp}, \mathcal{H}}^{\mathrm{DF}}(\epsilon) \leq m_{\mathrm{Exp}, \mathcal{H}}(\epsilon) \leq m_{\mathrm{Exp}, \mathcal{H}}^{\mathrm{DF}}(\epsilon/e),$$

*where $e \approx 2.718$ is Euler's number.*

The proof will make use of an equivalence between learners which attain low expected error and those which attain low error in the setting of *transductive learning*.

**Definition 27** *The (realizable) **transductive learning** model is defined by the following sequence of steps:*

1. *An adversary selects a collection of $n$ unlabeled datapoints $S = (x_1, \ldots, x_n) \in \mathcal{X}^n$, and a hypothesis $h^* \in \mathcal{H}$.*

2. *The unlabeled datapoints $S$ are displayed to the learner.*

3. *One datapoint $x_i$ is selected uniformly at random from $S$. The remaining datapoints are labeled by $h^*$ and displayed to the learner. That is, the learner receives $(x_j, h(x_j))_{j \neq i}$.*

4. *The learner is prompted to predict the label of $x_i$, namely $h^*(x_i)$.*

The **transductive error** incurred by a learner $A$ on a transductive learning instance $(S, h^*)$ is equal to its average prediction error over the uniformly random choice of $x_i$. That is,

$$L_{S, h^*}^{\mathrm{Trans}}(A) = \frac{1}{n} \sum_{i \in [n]} \ell\big(A(S_{-i}, h^*)(x_i), h^*(x_i)\big),$$

where $A(S_{-i}, h)$ denotes the output of $A$ on the sample consisting of all unlabeled datapoints in $S$ other than $x_i$, which are labeled by $h^*$. One can then define the **transductive error rate** of a learner $A$ as

$$\varepsilon_{A, \mathcal{H}}(n) = \max_{S \in \mathcal{X}^n, \, h \in \mathcal{H}} L_{S, h}^{\mathrm{Trans}}(A),$$

and similarly its **transductive sample complexity** as

$$m_{\mathrm{Trans}, A}(\delta) = \min\{m \in \mathbb{N} : \varepsilon_{A, \mathcal{H}}(m') < \delta, \, \forall m' \geq m\}.$$

Lastly, define the *transductive sample complexity* of a hypothesis class $\mathcal{H}$ as the pointwise minimal sample complexity attained by any of its learners, i.e.,

$$m_{\mathrm{Trans}, \mathcal{H}}(\epsilon) = \min_A m_{\mathrm{Trans}, A}(\epsilon).$$

We are now equipped to prove Lemma 26, and thus complete the proof of Theorem 9.

**Proof of Lemma 26** Certainly $m_{\mathrm{Exp},\mathcal{H}}^{\mathrm{DF}}(\epsilon) \leq m_{\mathrm{Exp},\mathcal{H}}(\epsilon)$, as any learner in the distribution-fixed model can elect to ignore the information of the unlabeled data distribution. For the second inequality, we argue that:

$$m_{\mathrm{Exp},\mathcal{H}}(\epsilon) \leq m_{\mathrm{Trans},\mathcal{H}}(\epsilon) \tag{1}$$

$$\leq m_{\mathrm{Exp},\mathcal{H}}^{\mathrm{DF}}(\epsilon/e). \tag{2}$$

Inequality (1) follows from the standard leave-one-out argument of transductive learning, which establishes that any learner $A$ incurring transductive error $\leq \epsilon$ on samples of size $n$ automatically incurs expected error $\leq \epsilon$ as well (Haussler et al., 1994). More explicitly, for any such learner $A$ and realizable distribution $D$ over $\mathcal{X} \times \mathcal{Y}$, one has:

$$
\begin{aligned}
\mathop{\mathbb{E}}_{S \sim D^m} L_D\big(A(S)\big) &= \mathop{\mathbb{E}}_{\substack{S \sim D^m \\ (x,y) \sim D}} \ell\big(A(S)(x), y\big) \\
&= \mathop{\mathbb{E}}_{S \sim D^{m+1}} \ell\big(A(S_{-(m+1)})(x_{m+1}), y_{m+1}\big) \\
&= \mathop{\mathbb{E}}_{S \sim D^{m+1}} \mathop{\mathbb{E}}_{i \in [m+1]} \ell\big(A(S_{-i})(x_i), y_i\big) \\
&\leq \sup_S \mathop{\mathbb{E}}_{i \in [m+1]} \ell\big(A(S_{-i})(x_i), y_i\big) \\
&\leq \epsilon.
\end{aligned}
$$

In pursuit of inequality (2), let $A$ be a distribution-family learner attaining expected error at most $\epsilon$ when trained on samples of size $\geq n$. We will design a (randomized) transductive learner $B$ attaining error at most $e \cdot \epsilon$ on samples of size $n$. For a training sample $S = (x_i, y_i)_{i \in [n]}$, let $S_{\mathcal{X}} = (x_i)_{i \in [n]}$ denote its unlabeled datapoints. Then $B$ acts as follows upon receiving a training set $S$ and test point $x$. If $x \in S$, $B$ simply returns the correct label for $x$, which was observed in $S$. Otherwise, $B$ generates a sample $T$ of $|S|$ many points drawn uniformly at random from $S$, and predicts $A(\mathrm{Unif}(S_{\mathcal{X}} \cup \{x\}), T)(x)$.

We now demonstrate that $B$ incurs error at most $e \cdot \epsilon$ on any transductive instance $S = (x_i, y_i)_{i \in [m]}$ with $m \geq n + 1$. Intuitively, this is due to the fact that $B$ mimics the performance of $A$ on i.i.d. data drawn from $\mathrm{Unif}(S)$, save for the fact that the test point $x$ must be excluded. Nevertheless, a sample of $|S|$ many points drawn i.i.d. from $S$ will happen to omit any given datapoint (such as $x$) with probability at least $\frac{1}{e}$. Thus, the lack of fidelity in $B$'s imitation of $A$ can at most inflate its error by a factor of $e$. More explicitly,

$$
\begin{aligned}
L_S^{\mathrm{Trans}}(B) &= \mathop{\mathbb{E}}_{i \in [m]} \ell\big(B(S_{-i})(x_i), y_i\big) \\
&= \mathop{\mathbb{E}}_{i \in [m]} \mathop{\mathbb{E}}_{T \sim \mathrm{Unif}(S_{-i})^{m-1}} \ell\Big(A\big(\mathrm{Unif}(S_{\mathcal{X}}), T\big)(x_i), y_i\Big) \\
&\leq e \cdot \mathop{\mathbb{E}}_{i \in [m]} \mathop{\mathbb{E}}_{T \sim \mathrm{Unif}(S)^{m-1}} \ell\Big(A\big(\mathrm{Unif}(S_{\mathcal{X}}), T\big)(x_i), y_i\Big) \\
&= e \cdot \mathop{\mathbb{E}}_{T \sim \mathrm{Unif}(S)^{m-1}} \mathop{\mathbb{E}}_{i \in [m]} \ell\Big(A\big(\mathrm{Unif}(S_{\mathcal{X}}), T\big)(x_i), y_i\Big) \\
&= e \cdot \mathop{\mathbb{E}}_{T \sim \mathrm{Unif}(S)^{m-1}} L_{\mathrm{Unif}(S)} A\big(\mathrm{Unif}(S_{\mathcal{X}}), T\big)
\end{aligned}
$$

$$\leq e \cdot \epsilon.$$

The third line makes use of the fact that a sample $T \sim \mathrm{Unif}(S)^{m-1}$ avoids any given point $x_i \in S$ with probability $\geq \frac{1}{e}$. (Recall that a transductive learning instead on a labeled dataset $|S| = m$ employs training sets of size $m - 1$.) In particular, let $f(m) = (1 - \frac{1}{m})^{m-1}$ denote the probability of a given $x_i \in S$ being avoided by $T \sim \mathrm{Unif}(S)^{m-1}$. First note that

$$\lim_{m \to \infty} f(m) = \lim_{m \to \infty} \left(1 - \frac{1}{m}\right)^{-1} \cdot \lim_{n \to \infty} \left(1 - \frac{1}{n}\right)^n = 1 \cdot \frac{1}{e} = \frac{1}{e}.$$

Furthermore,

$$\frac{\mathrm{d}}{\mathrm{d}x} f(x) = \frac{\left(\frac{x-1}{x}\right)^x \left(x \log \left(\frac{x-1}{x}\right) + 1\right)}{x - 1} \leq 0 \qquad \text{for } x > 1,$$

as $\left(\frac{x-1}{x}\right)^x > 0$, $x - 1 > 0$, and

$$x \log \left(\frac{x-1}{x}\right) + 1 = x \log \left(1 - \frac{1}{x}\right) + 1$$
$$\leq x \cdot -\frac{1}{x} + 1$$
$$= 0.$$

Thus $f(m)$ is weakly decreasing on $(1, \infty)$ and $f(m) \geq \frac{1}{e}$ for all $m \in \mathbb{N}$, as desired. ∎

