# OpenReview forum: "Proper Learnability and the Role of Unlabeled Data"
_algorithmiclearningtheory.org/ALT/2025/Conference — ALT 2025_

### Official Review · Reviewer_1Y2D · 2024-10-31
**Solid paper, leaning accept.**

**Rating:** 6
**Confidence:** 4

**Review:**

This paper studies the problem of PAC learning of multi-class function classes using proper learners. The paper shows that, unlike the improper learning case where the Daniely-Shalev-Shwartz (DS) dimension completely characterizes learnability, the learnability in the proper learning case can be more complex. In particular, the paper demonstrates the following:

1. If we are given the underlying distribution over $\mathcal{X}$ and assume the label is realizable, then the optimal sample complexity is achievable by a randomized proper learner (i.e., a Bayesian learner) up to a factor of $2$. This is achieved by representing the learning problem as a two-player zero-sum game and showing that the Bayesian learner attains an error up to only a factor of $2$ of the optimal error for any given prior over $\mathcal{H}$. The final Bayesian predictor then follows by invoking the minimax theorem.

2. The paper shows that there exists a class $\mathcal{H}$ such that its proper learnability is undecidable under ZFC. The construction follows from a simple manipulation of the EMX instance from Ben-David et al. (2019).

3. Finally, the paper shows that proper learnability is non-local and not monotone. This is achieved by augmenting a special function to the class constructed in point 2.

**Assessment:** This is clearly a solid paper and well within the scope of ALT. I find the results to be interesting and non-trivial. On the other hand, many of the proof ideas are not particularly surprising, e.g., the idea of expressing the proper learning problem as a zero-sum game has appeared in Darnstadt and Simon (2011), and the construction of hard instances follows too closely to Ben-David et al. (2019). However, as far as I know, these constructions are new.

Overall, I'm leaning toward accepting this paper. I also have some technical questions outlined below, which I hope the authors can address.

**Questions:**

1. (Major) I did not quite follow your proof of Lemma 25 (modulo the many typos), even after reading the reference in (Asilis et al., 2024b). Particularly, why is it true that "the transductive error incurred by B on S is equal to precisely the expected error incurred by A..."? From what I understand, the PAC error bound of $A$ is evaluated over $\text{Unif}(S)$, not the particular $x_{\text{test}}$. For example, if learner $A$ simply memorizes $S'$, its population error under $\text{Unif}(S)$ is upper bounded by $\frac{1}{n+1}$.

   I recommend the authors provide a step-by-step derivation of these claims, particularly writing explicitly the errors for $A$ and $B$ using mathematical formulas, not words.

2. (Minor) It seems your Theorem 13 holds only in expectation. What about high-probability sample complexity? (The repetition trick can only give an upper bound.)

3. (Minor) The hard instance in Proposition 20 uses uncountable labels. What if the label space is countable? Is it still impossible with a finite-dimensional measure to charaterizes the proper learnability?

4. (Minor) Can you provide any intuition as to why you believe Conjecture 16 is true?

**Paper Award:**

No

---

> ### Author Response · Authors · 2024-11-22
>
> We thank the reviewer for carefully reading the paper and for their thoughtful feedback.
>
> Regarding the questions:
>
> 1. This is indeed a bug in the existing approach (Lemma A.8  from Asilis et al. 2024b)– thank you for the correction! Fortunately, however, it can be resolved with a simpler proof. Let $A$ be a distribution-fixed learner attaining expected error $\leq \epsilon$ on samples of size $n$. Let $S = (x_1, y_1), \ldots, (x_{n}, y_{n})$ be a transductive learning instance, and let $D$ be the uniform distribution over the unlabeled points in $S$. Then, for transductive test point $x_i$, predict its label by running $A$ with the distribution $D$ as input and on $n$ training examples drawn uniformly at random from $S$, conditioned on not including the test point $x_i$. By classic coupon collector’s, $x_i$ is not in the sample with probability tending to $1 / e$ regardless of conditioning, so $A$’s error is multiplied by at most $e$ upon conditioning. We will correct this in the next version of the paper, and write out its proof very explicitly and precisely.
> 2. Correct, Theorem 13 only holds in expectation. As you mention, the repetition trick yields a randomized proper learner which is optimal up to logarithmic factors (yet which is no longer a distributional SRM learner). It is an interesting question whether distributional SRM learners are optimal in the high-probability regime; we have given this thought, but were unable to establish a result. We will mention this in the next version of the paper.
> 3. Yes, Proposition 20 uses uncountable labels, and this is essential to the current EMX-based approach. It is indeed a natural question whether this result can be realized with a countable label space, and we are not currently aware of whether or not this is possible. This is an interesting open question.
> 4. Conjecture 16 is motivated by the empirical observation that all examples of pathological hypothesis classes which are properly learnable (yet not by ERM) can be learned by SRM, to the best of our knowledge. Examples include the stochastic convex optimization problem in Section 4 of _Learnability, stability and uniform convergence (Shalev-Shwartz et al.)_, and the first Cantor class of _Optimal Learners for Multiclass Problems (Daniely and Shalev-Shwartz)_, augmented with the constant all-* behavior so as to be properly learnable. In short, we are not aware of any counter-example to the conjecture, and believe the community would benefit from the articulation of such a claim (and hopefully its resolution!).
>
>
> Thank you again for the detailed feedback.

---

> > ### Comment · Reviewer_1Y2D · 2024-12-01
> >
> > I thank the authors for the detailed response. The simplified proof appears to be correct to me. I have no issues accepting this paper.

---

### Official Review · Reviewer_xRN8 · 2024-11-03
**Review of the paper  *Proper Learnability and the Role of Unlabeled Data***

**Rating:** 7
**Confidence:** 2

**Review:**

The work done in the paper is mainly motivated by previous work of Daniely and Shalew-Shwartz showing that, in the multiclass setting, one can design a hypothesis class $\mathcal{H}$ which is PAC learnable, yet no learner for $\mathcal{H}$ can be *proper* (in other words, every learner for $\mathcal{H}$ is forced to output some predictor which is not an element of $\mathcal{H}$). This naturally rises the questions: *Which kind of supervised learning problems can be learned by some proper learners? And by what kinds of proper learners?* The focus of this paper is, in particular, on multiclass PAC learnability. Here, the second question is particularly important, as the seminal result of Brukhim, Carmon, Dinur, Moran and Yehudayoff, demonstrating that every hypothesis class with finite DS dimension is PAC learnable, employs a quite involved learning algorithm, which has very little in common to SRMs and other learning algorithms which are used in practice.

The main body of the paper can then be divided into two parts.

The first part of the paper established a positive result regarding the existence of a simple proper learner for **finite** multiclass problems (namely, for classes of hypotheses $\mathcal{H} \subseteq \mathcal{Y}^{\mathcal{X}}$, where both $\mathcal{X}$ and $\mathcal{Y}$ are finite), in the realizable setting and under the assumption that the learner has full knowledge of the probability distribution with respect to which samples are drawn. In particular, the authors show that such a proper learner can be expressed as a *distributional SRM*, a notion introduced in the paper generalizing that of classical SRM. As the authors themselves admit, the techniques used to prove such results are similar to those used in previous work by Darnstädt and Simon, although the goals of the two proofs are substantially different. I should point out that the proof currently given appears quite succinct, and I think that readability of the paper would benefit from a more detailed version: in particular, in view of the discussion of the result given on page 11, I suggest to explicitly point out where and how the distribution-fixed assumption is used, similarly to what is done in the paper by Darnstädt and Simon. This will help to make more evident which are the difficulties to extend the result to the classical setting and to better motivate their conjectures.

Notice that, in particular, the results by Darnstädt and Simon imply that, in the case of finite problems of binary classification (i.e.~hypothesis classes $\mathcal{H} \subseteq \lbrace 0, 1 \rbrace^{\mathcal{X}}$, where $\mathcal{X}$ is finite), knowledge of the distribution over the unlabeled data does not significantly help in the learning task. This same observation has been confirmed as valid in more general binary classification problems in various articles, a rather detailed list of which is provided by the authors. The authors complement their main result with a nice extension of such kind of findings to the multiclass setting. Indeed, they prove that:
- For every $\mathcal{X}, \mathcal{Y}$ and every hypothesis class $\mathcal{H} \subseteq \mathcal{Y}^{\mathcal{X}}$, $\mathcal{H}$ is PAC learnable in the *distribution-fixed model* (i.e.~when the learner is given full knowledge on the sampling distribution over $\mathcal{X}$) if and only if $\mathcal{H}$ is PAC learnable.

- Moreover, the sample complexity in the distribution-fixed model can only shrink up to a logarithmic factor with respect to that in the classical PAC model.

As the assumptions on the hypothesis class and the learning model of their main results are rather restrictive, the authors motivate the quest for milder hypotheses which still guarantee the existence of a proper learner. Indeed, a characterization of **proper** PAC learnability in the multiclass setting is an important theoretical open question (notice that, on the other hand, by the fundamental theorem of statistical learning, proper and improper PAC learnability coincide in the binary setting). The second part of the paper is then devoted to provide a series of negative results, substantiating the hardness of finding such a characterization. Using a simple yet smart reduction to the famous *EMX learning problem*, defined by Ben-David, Hrubes, Moran, Shpilka and Yahudayoff as an example of a learning framework in which learnability may be independent of ZFC, the authors prove that:

- There is a hypothesis class $\mathcal{H}$ (in which both the domain and the label set are uncountable) whose PAC learnability via some proper learner is independent of ZFC. In particular, proper learnability in the multiclass setting is not a *property of finite character*, in contrast with DS dimension (which characterizes *improper* learnability in the multiclass case) and all other known combinatorial dimension characterizing the various notions of learnability used in the literature.

- Proper PAC learnability in the multiclass setting is not a *local* property, namely there are hypothesis classes $\mathcal{H}, \mathcal{H}' \subseteq \mathcal{Y}^{\mathcal{X}}$ such that, for every finite $S \subseteq \mathcal{X}$, the restrictions of $\mathcal{H}$ and $\mathcal{H}'$ to $S$ coincide, yet $\mathcal{H}$ is properly learnable, while $\mathcal{H}'$ is not.

- Proper PAC learnability in the multiclass setting is not a *monotone* property, namely there are hypothesis classes $\mathcal{H}_0, \mathcal{H}_1, \mathcal{H}_2$ such that $\mathcal{H}_0$ is a proper subclass of $\mathcal{H}_1$, $\mathcal{H}_1$ is a proper subclass of $\mathcal{H}_2$, and yet only $\mathcal{H}_0$ and $\mathcal{H}_2$ are properly learnable.

Notice, however, that a similar reduction (although technically more complicated) to EMX learning has been recently used in the paper *Bandit learnability can be undecidable* by Hanneke and Yang: I think that the authors should include such a reference, as it gives another natural example of a learning framework with problems whose learnability is independent of ZFC.

In summary, this is a list of what are, in my opinion, the strong and weak aspects of this paper:

**PROS**

- The results of the paper are well-motivated, as well as their position with respect to the state of the art.

- The review of the related work concerning the first part of the paper is adequate.

- Theorem 9 significantly extends to the multiclass setting the refutation of the power of unlabeled data in absence of further assumptions, at least from the worst-case perspective, nicely complementing the state-of-the-art knowledge.

- Although technically simple, the corollaries obtained by means of the reduction to EMX learning provide strong evidence of the hardness of finding a characterization of proper multiclass PAC learnability, showing clearly that a brand new approach, substantially different to the existing combinatorial dimensions, is needed to tackle this question.

**CONS**

- Readability and clarity of the proof of the main theorem (Theorem 13) may be improved, as well as the one of the discussion immediately following (on page 11).

- The applicability of Theorem 13, as it currently is, are quite limited. It would be nice to have some (even partial) extension at least to classes of finite DS dimension, similarly to the one given by Darnstädt, Simon and Szörényi (*Unlabeled Data Does Provably Help*).

- The review of related work related to the second part of the paper can be improved. Besides the missing reference already mentioned, the authors should consider to briefly mention some of the results concerning the *algorithmic* decidability of PAC learning, whose investigation arises precisely as a consequence of the seminal result by Ben-David, Hrubes, Moran, Shpilka and Yahudayoff. Indeed, although not directly related, the line of research mentioned above shares with this paper the motivation of finding **simple** learners, which, in that case, are identified with learning *algorithms* (rather then simply *functions*), in the strict sense of computability theory. A large part of the literature on such problem is briefly reviewed, for example, in the paper *From undecidability of non-triviality and finiteness to undecidability of learnability* by Caro. In particular, the paper *Find a witness or shatter: the landscape of computable PAC learning* by Delle Rose, Kozachinskiy, Rojas and Steifer, solves a problem left open in *On characterizations of learnability with computable learners* by Sterkenburg providing a characterization of those classes which are **properly** PAC learnable via some computable learner by means of a relaxation of the notion of ERM: this example seems to fit the context of the paper particularly well.

- To the best of my knowledge, the novelty of the paper from the technical point of view is quite limited (besides the introduction of distributional SRMs, which, on the other hand, are only really used in the form of Bayesian learners).

**Paper Award:**

No

---

> ### Author Response · Authors · 2024-11-22
>
> We thank the reviewer for carefully reading the paper and for their thorough and thoughtful feedback.
>
> Regarding the proof of Theorem 13, we agree that the paper would benefit from additional detail and discussion of where the distribution-fixed requirement is being used (in particular, in the definition of the payoff function for the game). We will update the paper to include such detail.
>
> Regarding the reference to the paper _Bandit learnability can be undecidable_, we agree that this certainly deserves mention, and will update the paper accordingly. More broadly, we thank the reviewer for mentioning several related works on the algorithmic decidability of PAC learning, which is certainly of relevance to our results on logical undecidability and to our search for simple learners. We will update the paper to include all the mentioned references (and some others which also appear to be relevant, such as _On Learnability with Computable Learners_.)
>
> We agree that it would be desirable to generalize Theorem 13 to more general domains $\mathcal{X}$ and label sets $\mathcal{Y}$, as the finiteness requirement is quite restrictive. However, the current proof technique, which is based upon appealing to the minimax theorem, does not seem to generalize easily to more general spaces. (I.e., one faces the issue of considering the sigma algebra upon $\mathcal{X}$, of topologizing the collection of probability measures over $\mathcal{X}$, etc.) We will add a remark mentioning such complexities in the next version of the paper.
>
> Thank you again for the detailed feedback!

---

### Official Review · Reviewer_DaG9 · 2024-11-11
**Review of "Proper Learnability and the Role of Unlabeled Data"**

**Rating:** 7
**Confidence:** 3

**Review:**

This paper studies multiclass learning problems with a specific focus on understanding how unlabelled data helps with \emph{proper learning}. The authors study the ``distribution fixed'' PAC model where the learning algorithm is given access to unlabelled data via a full description of the marginal distribution $\mathcal{D}$ on the instance space $\mathcal{X}$ (and hence can generate as many i.i.d. as desired.) This model still retains a worst case perspective by evaluating on the hardest $\mathcal{D}$ and hardest target function $f \in \mathcal{F}$. As a first result, the authors show that the sample complexity of this setting and the standard PAC setting are essentially equivalent up to a logarithmic factor in the failure probability, demonstrating that unlabelled data cannot help much for the worst distribution. The authors then go on to show that in the distribution fixed setting, an optimal learning strategy (up to a constant factor of 2) is achieved by a \emph{randomized proper} learning strategy whenever the instance space $\mathcal{X}$ and label space $\mathcal{Y}$ are finite. The authors then consider the standard multiclass PAC setting and show several negative results regarding characterizing proper learnability by proving that: 1) proper learning can be logically undecidable 2) that it is not monotone property of function classes (wrt to addition/removal of functions) and 3) it is not a local property.

I generally find the results in the paper to be quite interesting. In particular, the result which demonstrates that there exists a randomized proper learner that is optimal in expectation up to a factor of 2 is interesting because this result is somewhat in contrast to an implication of known result in the standard multiclass PAC setting (considered with finite $\mathcal{X}$ and $\mathcal{Y}$) that says any proper learner suffers constant expected error though the comparison is not perfect since the upper bound in this paper allows for randomization. I am a bit confused though why the authors refer to these as randomized proper strategies (and in fact drop the randomized part in many places in the paper) since they are mixtures of functions in the class which can be much more expressive than the functions themselves. This can give off the false impression that these learners are truly proper especially when the \emph{randomized} qualifier is dropped. It would also be interesting to see a discussion regarding the specific case of standard binary classification where we known simpler optimal algorithms (up to constants) based on taking majority votes of ERMs (which to be fair is not the same as randomized proper) and whether it is possible to prove similar results here. Finally, while I find the final set of results around undecidability to be interesting, I feel that they are less of a contribution since they are a very straight forward consequence of existing results.

**Paper Award:**

No

---

> ### Author Response · Authors · 2024-11-22
>
> We thank the reviewer for carefully reading the paper and for their thoughtful feedback.
>
> Regarding the comment about randomized proper strategies, we believe that it is appropriate to refer to them as “randomized proper” owing to the fact that for any crystallization of the learner’s randomness, it outputs a hypothesis in the underlying class $\mathcal{H}$. Furthermore, in many natural cases, a randomized learner can be derandomized by reserving a fraction of training data to serve as a source of randomness (e.g., using the machinery of Bernoulli factories). We agree, however, that dropping the qualifier “randomized” can give a false impression, and will edit the paper to clarify this point.
>
> We will also add discussion of the Majority-of-Three learner from the binary case, which we agree is relevant in understanding the power of proper learning for classification. Regarding whether it is possible to prove similar results in the general multiclass setting, we note that some concurrent work appears to have explored this question and established both positive and negative results (“Understanding Aggregations of Proper Learners in Multiclass Classification” by Asilis, Hogsgaard, and Velegkas 2024). We will comment on this work and how it relates to our results as well.
>
> Thank you again for the detailed feedback!

---

### Official Review · Reviewer_2CNW · 2024-11-11
**conceptually interesting paper on proper learnability**

**Rating:** 7
**Confidence:** 4

**Review:**

The paper “Proper Learnability and the Role of Unlabeled Data” examines the conditions under which proper learning can be optimal (as opposed to when an improper learner is needed for optimality) and also studies whether it can be simply charactarized.

The main results of the paper are as follows:
- in the distribution dependent pac model, optimal proper learners exist
- the efficiency loss from distribution-dependent to classical PAC is at most logarithmic
- impossibility results for multiclass proper learning, going through the EMX work of Ben-David et al., are given.

These results are conceptually interesting and add to the picture of proper vs improper learnability that has been analyzed by many people including Simon, Danielly, Hanneke, etc.  Some of these results, however, are not very complicated and goes through known avenues of attack, eg via results for the transductive setting or via techniques developed by Daniely and Shalev-Shawrtz. The latter results illustrate the difficulty of characterizing proper learnability in many settings.

Despite these reservations, I think the conceptual contributions warrant acceptance. The paper is very well-written and easy to follow.

**Paper Award:**

No

---

> ### Author Response · Authors · 2024-11-22
>
> We thank the reviewer for carefully reading the paper and for their thoughtful feedback. We appreciate the positive reception to the paper, and agree that our work stands on the shoulders of Simon, Daniely, Hanneke, and others.

---

### Author Rebuttal · Authors · 2024-11-22

We thank all reviewers for the detailed feedback on our paper. We have responded to each reviewer individually. Thank you again!

---

### Meta-Review · Area_Chair_oBCt · 2024-12-14

**Recommendation:** Accept
**Confidence:** 5

**Metareview:**

This paper investigates proper PAC learning. With many recent results in PAC-type learnability being derived through improper learners (eg the recent characterization of multi-class learnability by the DS dimension) the role of proper learners for (optimal) learnability is a natural and intriguing avenue to explore.

The work starts by establishing that in the setting of known marginals, optimal proper learners exist. Further, it is shown that the sample complexities of known marginal PAC versus general PAC learning do not differ beyond logarithmic factors, which strengthens results on the lack of benefits of unlabeled data for supervised PAC learning. Finally, the submission presents a reduction of EMX learnability to proper multiclass learnability, and thus shows that proper multiclass learnability inherits the potential of independence of ZFC from EMX.

The reviewers all appreciated the submission and some initial concerns were clarified through the author response. The reviewers also pointed to a variety of minor issues with the write-up at submission time (typos, unclarities in terminology used, references), which the authors are asked to carefully address for publication.

**Paper Award:**

No